# NK cell exhaustion in Wilson's disease revealed by single-cell RNA sequencing predicts the prognosis of cholecystitis

Yong Jin[1†], Jiayu Xing[1†], Chenyu Dai[2†], Lei Jin[3,4], Wanying Zhang[1], Qianqian Tao[3,4], Mei Hou[1], Ziyi Li[3,4], Wen Yang[1,5,6*], Qiyu Feng[1*], Hongyang Wang[1,5,6*], Qingsheng Yu[2,3*]

[1]Cancer Research Center, The First Affiliated Hospital of USTC, Division of Life Sciences and Medicine, University of Science and Technology of China, Hefei, China; [2]Department of Cadre Cardiology, The First Affiliated Hospital of Anhui University of Chinese Medicine, Hefei, China; [3]Department of General Surgery, The First Affiliated Hospital of Anhui University of Chinese Medicine, Hefei, China; [4]Institute of Chinese Medicine Surgery, Anhui Academy of Chinese Medicine, Hefei, China; [5]International Co-operation Laboratory on Signal Transduction, Eastern Hepatobiliary Surgery Hospital, Second Military Medical University, Shanghai, China; [6]National Center for Liver Cancer, Second Military Medical University, Shanghai, China

*For correspondence:
woodeasy66@hotmail.com (WY);
qiyufeng@ustc.edu.cn (QF);
hywangk@vip.sina.com (HW);
qsy6312@163.com (QY)

†These authors contributed equally to this work

Competing interest: The authors declare that no competing interests exist.

## eLife Assessment

This study presents **valuable** findings, based on **solid** methods, to link metabolic dysfunction in Wilson's disease to immune cell dysregulation and poor cholecystitis outcomes. The integration of clinical data and single-cell analyses highlights NK cell exhaustion as a key factor, offering insights with potential therapeutic implications. The work will be of interest to colleagues in inflammatory and metabolic diseases.

**Abstract** Metabolic abnormalities associated with liver disease have a significant impact on the risk and prognosis of cholecystitis. However, the underlying mechanism remains to be elucidated. Here, we investigated this issue using Wilson's disease (WD) as a model, which is a genetic disorder characterized by impaired mitochondrial function and copper metabolism. Our retrospective clinical study found that WD patients have a significantly higher incidence of cholecystitis and a poorer prognosis. The hepatic immune cell landscape using single-cell RNA sequencing showed that the tissue immune microenvironment is altered in WD, mainly a major change in the constitution and function of the innate immune system. Exhaustion of natural killer (NK) cells is the fundamental factor, supported by the upregulated expression of inhibitory receptors and the downregulated expression of cytotoxic molecules, which was verified in clinical samples. Further bioinformatic analysis confirmed a positive correlation between NK cell exhaustion and poor prognosis in cholecystitis and other inflammatory diseases. The study demonstrated dysfunction of liver immune cells triggered by specific metabolic abnormalities in WD, with a focus on the correlation between NK cell exhaustion and poor healing of cholecystitis, providing new insights into the improvement of inflammatory diseases by assessing immune cell function.

## Introduction

Cholecystitis refers to a group of inflammatory diseases with different etiologies and clinical courses (*Elwood, 2008*). While the corresponding pathological mechanisms of cholecystitis are complicated, systemic inflammation is widely recognized as a primary catalyst of cholecystitis pathology (*Huffman and Schenker, 2010*; *Markaki et al., 2021*; *Laurila et al., 2005*; *Fu et al., 2022*; *Abu-Sbeih et al., 2019*). Cholecystectomy is typically the preferred treatment option for patients with severe cholecystitis and gallstones, whereas medication is prescribed for those with mildly symptomatic cholecystitis or calculous cholecystitis. However, a substantial percentage of patients face the possibility of complications or recurrence (*Loozen et al., 2017*; *van Dijk et al., 2016*). Compelling evidence supports that the proportion, gene expression profile, and functional status of immune cells hold significant prognostic value (*Buonacera et al., 2022*; *McKinney et al., 2010*; *McKinney et al., 2015*; *Sun et al., 2021*; *Fridman et al., 2017*; *Wouters and Nelson, 2018*). Therefore, it is critical to comprehensively comprehend the immune cell dysfunction linked with cholecystitis, its corresponding altered characteristics, and their prognostic implications.

Natural killer (NK) cells are the major innate lymphocyte subset and are found in most organs (*Abel et al., 2018*; *Crinier et al., 2020*). They produce lytic granules that eliminate infected or cancerous cells, coordinate the inflammation process, form immunological memory, and modulate antigen-presenting cell function in inflammatory diseases (*Highton et al., 2021*; *Parisi et al., 2017*; *Earls and Lee, 2020*). In the context of tumors and chronic infections, NK cells exhibit an exhausted phenotype, termed NK cell exhaustion, which is characterized by downregulated expression of activating receptors (e.g. NKG2D, FCGR3A), upregulated expression of inhibitory receptors (e.g. KLRC1, PD-1, TIGIT, Tim-3), decreased production of effector cytokines (e.g. IFNγ, GZMB), and impaired cytolytic activity (*Bi and Tian, 2017*; *Zhang et al., 2018*). Recently, blockade of KLRC1 and TIGIT has been shown to reverse NK cell exhaustion and enhance the therapeutic effects of anti-tumor and anti-infective immunotherapies (*Zhang et al., 2018*; *Liu et al., 2023*; *Zhang et al., 2019b*). In addition, NK cell exhaustion has been reported to predict poor prognosis in hepatocellular carcinoma and acute myeloid leukemia; however, the correlation of NK cell exhaustion with prognosis in cholecystitis and other inflammatory diseases remains largely unexplored (*Sun et al., 2019*; *Liu et al., 2022*; *Sun et al., 2017*).

The liver plays a critical role in immune defense. It contains a substantial population of diverse immune cells, particularly NK cells (*Crispe, 2009*; *Kubes and Jenne, 2018*; *Racanelli and Rehermann, 2006*). The liver modulates these cells to mount immune responses upon encountering specific inflammatory triggers (*Kubes and Jenne, 2018*; *Hardy et al., 2016*). Furthermore, liver pathologies affect mesenchymal immune cell function and influence systemic inflammatory responses, impacting the development and outcome of inflammation-associated diseases such as cholecystitis (*Armstrong et al., 2014*; *Matyas et al., 2021*; *Knab et al., 2014*). Wilson's disease (WD) is a rare genetic disorder caused by mutations in the gene encoding the transmembrane copper-transporting P-type ATPase protein (*Bandmann et al., 2015*). It leads to deficits in mitochondrial structure and function, as well as specific alterations in liver metabolism, resulting in a variety of liver diseases. Additionally, WD is commonly associated with inflammatory conditions in the clinic (*Huster et al., 2006*; *Medici et al., 2013*; *Huster, 2014*; *Aggarwal and Bhatt, 2020*). Thus, WD, with its unique etiology, serves as an excellent model for exploring the regulation of immune cell function by hepatic metabolic abnormalities.

In this retrospective clinical study of over 600 WD patients, it was found that they had a significantly higher cholecystitis incidence than the general population. This, in turn, is associated with a poor prognosis for cholecystitis in these patients. A thorough examination of the immune cell landscape in the hepatic mesenchymal stromal microenvironment of WD patients was performed using single-cell RNA sequencing (scRNA-seq). Specifically, there were significant changes in the composition and function of the innate immune system, including an increase of mononuclear phagocytes (MPs) and NK cells, as well as enhanced pro-inflammatory characteristics. Additionally, there was an enhancement in the biological processes of antigen presentation, activation of immune response, and activation of lymphocytes. One key event among these changes was the exhaustion of NK cells, as demonstrated by an increase in the expression of inhibitory receptors KLRC1 and TIGIT and a decrease in the expression of cytotoxic molecules. Multiplex immunohistochemistry (mIHC) and flow cytometry techniques confirmed the exhaustion of NK cells in both clinical liver tissue and peripheral blood samples. Finally, our bioinformatics analysis has confirmed a positive correlation between NK cell exhaustion and poor

prognosis in cholecystitis and other inflammatory diseases. Our study has demonstrated abnormal liver mesenchymal immune cell function triggered by specific metabolic dysfunction in WD, focusing on NK cell exhaustion and its correlation with poor healing of cholecystitis. These findings provide foundation for further studies of effect of hepatocytes metabolic changes on immune cell function and insights into improvement of inflammatory diseases by assessing immune cell function.

## Results

### WD causes poor prognosis of cholecystitis

WD is a genetic disorder resulting from mutations in the *ATP7B* gene, exhibiting significant genetic heterogeneity with over 600 mutations or polymorphisms (*Cheng et al., 2017*). Through sequencing DNA samples from 20 patients with WD, we found that c. 2333G>T (p. R778L) accounted for the majority (65%) of the three most common mutations in Chinese, while c. 2804C>T (p. T935M) and c. 2975C>T (p. P992L) were not detected (*Figure 1a*; *Yang et al., 2021*). In hepatocytes, ATP7B deficiency in the trans-Golgi network (TGN) leads to abnormalities in mitochondrial function and copper metabolism, and ultimately to a wide range of liver diseases (*Figure 1b*; *Członkowska et al., 2018*; *Roberts et al., 2008*; *Zischka and Lichtmannegger, 2014*; *Muchenditsi et al., 2021*). Typical clinical manifestations are liver fibrosis and Kayser-Fleischer ring in the cornea that caused by Cu accumulation in the cornea (*Figure 1c and d*; *Członkowska et al., 2018*). Additionally, a lower total bilirubin and metabolic symptoms of WD patients, a lower risk of hyperglycemia (Cohort 1 and Cohort 2), are observed (*Figure 1e*, *Figure 1—figure supplement 1a*). Mitochondrial stress and glycolysis assays revealed that knockout of the *ATP7B* (ATP7B-KO) in human hepatocyte cell lines WRL 68 significantly impaired the basal oxygen consumption rate (OCR) and capacity of ATP production, while the extracellular acidification rate (ECAR) of glycolysis and glycolysis capacity remained unchanged (*Figure 1—figure supplement 1b and c*). These results demonstrate that WD is a disease characterized by abnormal hepatocyte metabolism.

Interestingly, chronic cholecystitis emerged as the most common complication, with a notably higher incidence in WD patients (49.51%) compared to HBV patients (9.46%) in Cohort 2 (*Figure 1e*). Despite the comparable incidence of gallbladder in the two groups and the significantly reduced plasma concentration of total bilirubin in WD patients (Cohort 1 and Cohort 2), this finding holds (*Figure 1e*). Moreover, in cholecystitis patients, the proportion of cases with abnormal index was significantly higher when combined with WD, which included white blood cell (WBC), neutrophil (NEU), and lymphocyte (LYM) counts (Cohort 3, *Figure 1e*). Overall, these findings suggest that cholecystitis outcomes are negatively affected by WD.

### Single-cell profiling of non-parenchymal cells in the liver of WD patients

The hepatic mesenchymal cells from three WD patients and three liver hemangioma patients were analyzed using the 10x Genomics platform for scRNA-seq (*Figure 2a*) to better understand the complicated immune cell abnormalities in WD patients. 57,384 high-quality single transcriptomes were generated, with 30,068 cells in the group of WD patients (CASE) and 27,316 in the group of liver hemangioma patients (CON). We validated 26 principal components via PCA and substantiated the grouping through UMAP dimensionality reduction cluster graph (*Figure 2b and c*, *Figure 2—figure supplement 1a*). We then utilized individual cell expression profiles and reference to cell marker gene expression to implement an automated annotation method, which was further complemented by manual review and fine-tuning. This initial annotation process resulted in the classification and display of 12 significant cell clusters. The UMAP algorithm was used to segregate and display the clusters. The study identified various cell types, including hepatocytes (*He et al., 2020*), cholangiocytes (*He et al., 2020*), endothelial cells (ECs) (*Li et al., 2021*; *Sathe et al., 2020*; *Wang et al., 2020d*), hepatic stellate cells (HepSCs) (*Bhattacharjee et al., 2021*), proliferating cells (*Sathe et al., 2020*; *Dumas et al., 2020*; *Durante et al., 2020*), B cells (*Gong et al., 2021*; *Wang et al., 2020c*), plasma cells (*Durante et al., 2020*; *Gong et al., 2021*), T and NK cells (*Gong et al., 2021*; *Zeng et al., 2019*; *Zhang et al., 2023*), neutrophils (*Wu et al., 2021*), mast cells (*Sathe et al., 2020*; *Zeng et al., 2019*; *Wu et al., 2021*), MPs (*Ramachandran et al., 2019*), and plasmacytoid dendritic cells (pDCs) (*Sathe et al., 2020*; *Pombo Antunes et al., 2021*; *Figure 2d, e* and *Figure 2—figure supplement 1b, c*). *Figure 2—figure supplement 1d* exhibited the top 10 differently expressed genes (DEGs) for each cell type.

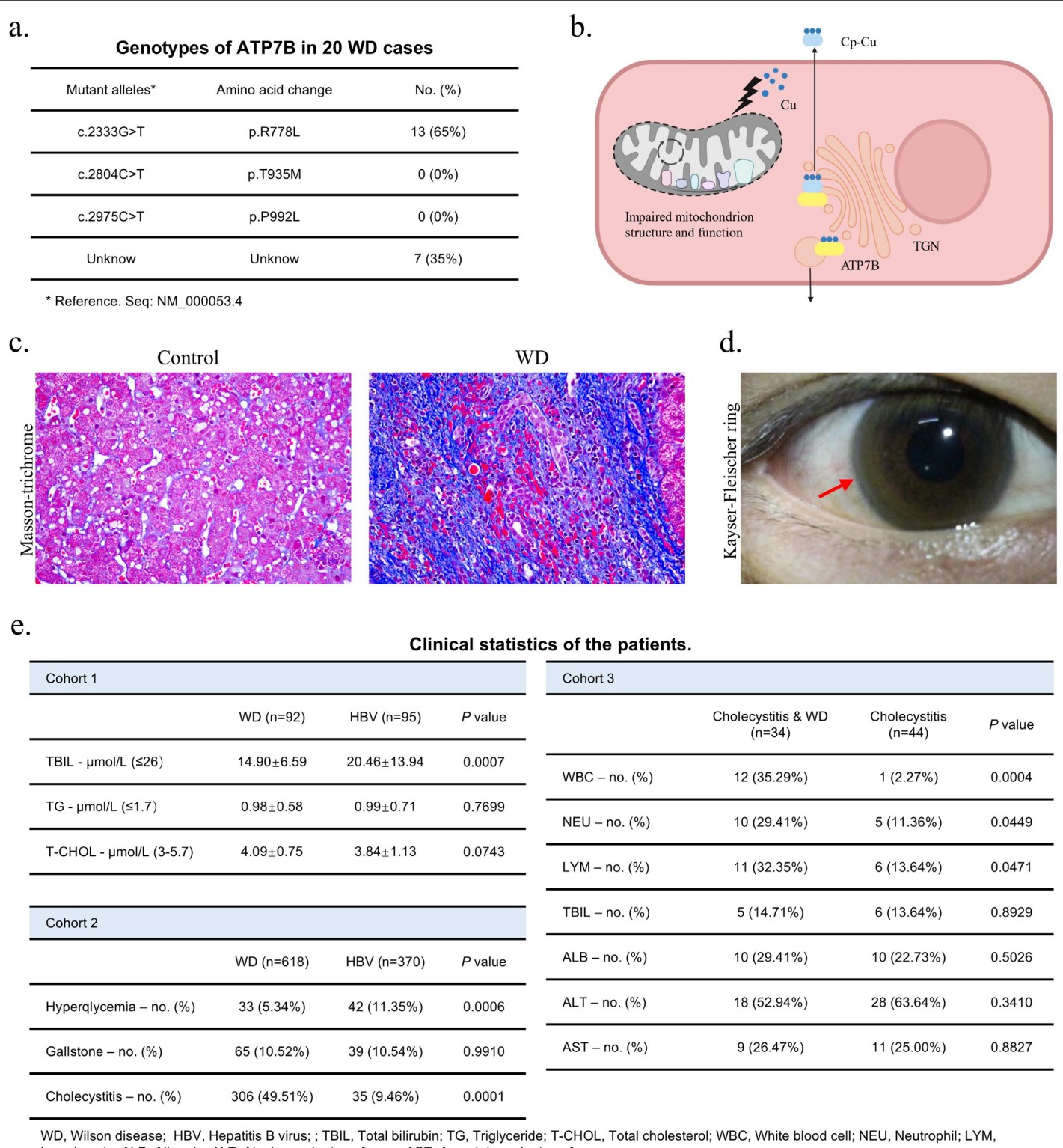

**Figure 1.** Clinical manifestation of Wilson's disease (WD) patients and WD causes poor prognosis of cholecystitis. (**a**) DNA sequencing result of 20 WD patients was summarized in the graph. (**b**) The schematic diagram illustrates copper transport disturbances caused by ATP7B mutations, along with structural and functional impairments in mitochondria. Created with BioRender.com. (**c**) Masson-trichrome staining of liver tissues of patients with WD and control. (**d**) The picture of Kayser-Fleischer (KF) ring in the cornea of a patient with WD, captured during clinical examination. (**e**) The table of clinical statistics of patients in three cohorts. Data are mean ± SD. Unpaired two-tailed t-test and chi-square test.

*Figure 1 continued on next page*

*Figure 1 continued*

The online version of this article includes the following figure supplement(s) for figure 1:

**Figure supplement 1.** *ATP7B* mutation causes metabolic abnormalities.

Based on the initial annotation, there were notable variations in the quantities and ratios of different cell types between CASE and CON. Our study observed that T and NK lineages were the primary immune cell population among all the examined samples, accounting for 56.54% in CASE and 73.55% in CON, followed by MPs which constituted 17.33% in CASE and 9.97% in CON (*Figure 2f*, *Figure 2—figure supplement 1e*, and *Supplementary file 1a*). The significant increase in the proportion of MPs and decrease in T and NK cells suggest that MPs likely dominate the immune microenvironment remodeling in WD patients, while lymphocytes play a minor role. B cell proportions (4.53% in CASE and 2.27% in CON), plasma cell proportions (2.36% in CASE and 0.42% in CON), and pDC proportions (0.42% in CASE and 0.16% in CON) were elevated, while neutrophil proportions (3.53% in CASE and 6.45% in CON) declined. The proportion of remaining cells is shown in *Supplementary file 1a*.

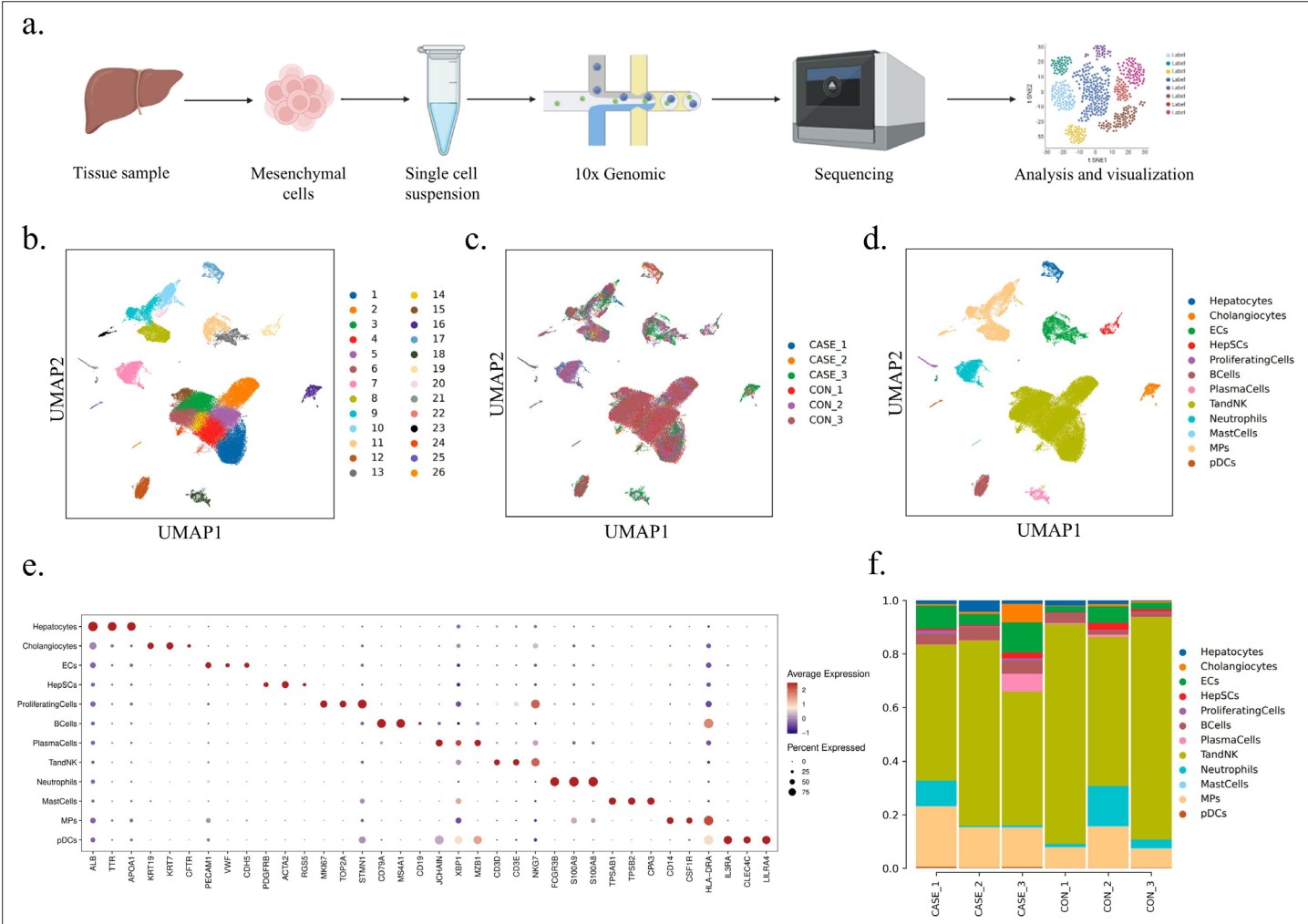

**Figure 2.** Single-cell profiling of non-parenchymal cells in the liver of Wilson's disease (WD) patients. (**a**) The workflow of single-cell RNA sequencing (scRNA-seq). Created with BioRender.com. (**b**) UMAP visualization of 26 principal components of all the single cells. (**c**) UMAP visualization of 26 principal components by samples. (**d**) UMAP visualization of 12 main cell types. (**e**) The dot plot showing the expression of top 3 marker genes in 12 main cell types. (**f**) The bar chart showing the proportion of 12 main cell types in each sample.

The online version of this article includes the following figure supplement(s) for figure 2:

**Figure supplement 1.** The proportion and differently expressed genes (DEGs) of each main cell type.

## An overview of the immune microenvironment in WD patients

To demonstrate dissimilarities in the composition of the immune microenvironment between the two groups of individuals in a more detailed manner, a secondary subtype clustering and annotation process was applied to MPs and T and NK cells (*Figure 3a, b* and *Figure 3—figure supplement 1a, b*). MPs are myeloid immune cells that serve phagocytic and antigen-presenting functions. These cells, which include monocytes, macrophages, and dendritic cells, play an essential role in determining the nature of the immune response (*Strauss et al., 2015*). A total of 6962 MPs were identified (4439 in CASE and 2523 in CON), and they were subdivided into seven clusters through annotation (*Figure 3a*, *Figure 3—figure supplement 1a*, and *Supplementary file 1a*). Cell marker genes for annotating cell subtypes were depicted in *Figure 3—figure supplement 1c and d*, and *Figure 3—figure supplement 1e* demonstrated the top 10 DEGs between the two groups (*Sun et al., 2021*; *Sathe et al., 2020*; *Wang et al., 2020d*; *Dumas et al., 2020*; *Durante et al., 2020*; *Zeng et al., 2019*; *Zhang et al., 2023*; *Wu et al., 2021*; *Ramachandran et al., 2019*; *Pombo Antunes et al., 2021*; *Singh et al., 2019*; *Halbritter et al., 2019*; *Bassez et al., 2021*; *Popescu et al., 2019*; *Chen et al., 2020*; *Wang et al., 2020b*). The DEGs were related to various cell types, including proliferating cells, macrophages, monocytes, mature dendritic cells (MatureDCs), conventional type 1 dendritic cells (cDC1), conventional type 2 dendritic cells (cDC2), and Kupffer cells (KCs). In CASE, there was a significant increase in both macrophages (25.03% in CASE versus 14.33% in CON) and KCs, the resident macrophages in the liver (26.03% in CASE versus 22.96% in CON), suggesting their involvement in promoting hepatic inflammation in metabolic liver disease (*Figure 3a*; *Dixon et al., 2013*). By contrast, the percentages of monocytes (35.45% in CASE and 41.78% in CON), cDC1 (2.14% in CASE and 5.99% in CON), and cDC2 (10.39% in CASE and 14.13% in CON) were decreased in CASE (*Figure 3a*).

Seventeen clusters were obtained through reclustering for the largest population of T and NK cells, totaling 34,339 cells (15,403 cells in CASE and 18,936 in CON) (*Gong et al., 2021*; *Wu et al., 2021*; *Cella et al., 2019*; *Elmentaite et al., 2021*; *Liu et al., 2020*; *Wang et al., 2020a*; *Tirosh et al., 2016*; *Jaeger et al., 2021*; *Depuydt et al., 2020*; *Andor et al., 2019*; *Heming et al., 2021*; *Xin et al., 2022*; *Zhao et al., 2020*; *Williams et al., 2021*; *Madissoon et al., 2019*; *Zhang et al., 2020*; *Fernandez et al., 2019*; *Luoma et al., 2020*). The clusters included ILC3_IL1R1, NKT_GNLY, NKT_IFNG, NKT_KLRG1, NK_FCER1G, NK_KLRC1, NKT_XCL2, NK_FCGR3A, CD4NaiveT_CCR7, CD4Tmem_CD40LG, CD4Treg_CTLA4, CD8MAIT_KLRB1, CD8MAIT_SLC4A10, CD8Teff_GZMH, CD8Teff_GZMK, CD8Teff_ISG15, and CD8Teff_MT1X (*Figure 3b*; *Figure 3—figure supplement 1b*; *Supplementary file 1a*). The proportion of NK cells (40.97% and 17.07% in CASE and CON, respectively) and CD4Treg (0.92% and 0.31% in CASE and CON, respectively) was significantly increased in CASE. On the other hand, a decrease of NKT cells (28.72% in CASE and 35.26% in CON), CD8MAIT (7.76% in CASE and 22.14% in CON), and CD8Teff (14.95% in CASE and 17.88% in CON) was observed in CASE. CD4 regulatory T cells are thought to inhibit T cell activation in autoimmune and virus-related liver diseases, indicating a correlation between elevated CD4 regulatory T cells and reduced CD8 T cells in WD patients (*Zhang et al., 2016*). The distribution of other types of cells between the two groups is shown in *Figure 3b* and *Supplementary file 1a*.

To systematically evaluate the key functional alteration, we analyzed the top 20 DEGs in all immune cell populations. We found that the expression of genes encoding complement C1q (*C1QA*, *C1QB*, and *C1QC*) and *HLA-DRA* was significantly upregulated in CASE. Conversely, the expression of genes encoding heat shock protein (*HSPA1A*, *HSPA1B*, *HSPM1*, *HSPA6*, *HSP90AB1*, *HSPE1*, and *HSPD1*) was markedly downregulated in CASE (*Figure 3c*). The Gene Ontology (GO) analysis results demonstrate that WD enhances the abilities of antigen-presenting cells to activate immune response and related lymphocytes, through upregulating antigen processing and presentation, activation of immune response, and positive regulation of lymphocyte activation. However, heat shock proteins delivering antigens may be responsible for the downregulation of lymphocytes differentiation and T cell activation (*Shevtsov and Multhoff, 2016*). Next, we analyzed the cellular communication network using CellChat. The primary signal senders and receivers were shown, with their respective strength. Our findings reveal that CD8Teff, KCs, NKT, CD8MAIT, monocytes, cDC2, and macrophage were the primary participants in intercellular communication, with the high value in both incoming and outgoing interaction strength (*Figure 3e*). Furthermore, the incoming interaction strength of CD8Teff and NK cells was higher in CASE, suggesting the regulation of biological process by extracellular signals, while neutrophils and monocytes exhibited the lower interaction strength in both outgoing

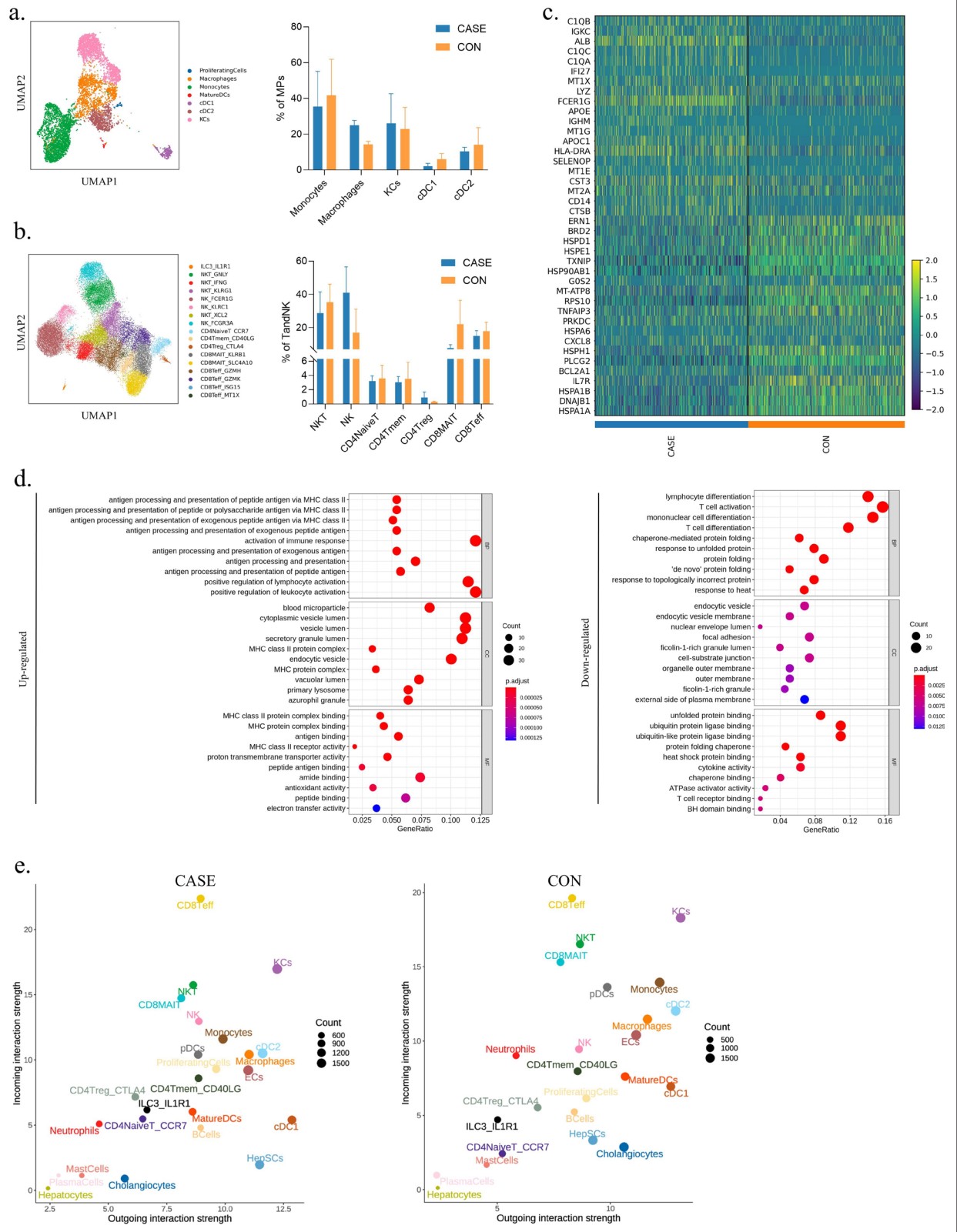

**Figure 3.** An overview of the immune microenvironment in Wilson's disease (WD) patients. (**a**) UMAP visualization of mononuclear phagocytes (MPs) colored and labeled by cell subtypes (left). The bar chart showing the average percentage of different subtypes in CASE (n=3) and CON (n=3) (right). Data are mean ± SD. (**b**) UMAP visualization of T and NK cells colored and labeled by cell subtypes (left). The bar chart showing the average percentage of different subtypes in CASE (n=3) and CON (n=3) (right). Data are mean ± SD. (**c**) The heatmap showing the top 20 differently expressed genes (DEGs)

*Figure 3 continued on next page*

*Figure 3 continued*

for all immune cells. (**d**) The dot plot showing Gene Ontology (GO) enrichment on DEGs of all immune cells between CASE and CON. BP, biological processes; CC, cellular components; MF, molecular function. (**e**) The scatter showing the incoming and outgoing interaction strength of all cell subtypes in CASE and CON.

The online version of this article includes the following figure supplement(s) for figure 3:

**Figure supplement 1.** The gene expression characteristics of each cell subtype.

and incoming (*Figure 3e*). Collectively, these results demonstrate changes in the composition and effects of innate immunity in the liver of WD patients.

## The dysfunction of main immune cells in WD patients

To obtain a detail analysis of function comparison in immune cells between two groups, we scored and evaluated the core functional signature of major cell types. We further clustered macrophages, which accounted for 14.33% of the MPs in CON and increased to 25.03% in CASE, resulting in the identification of four distinct subtypes (*Figure 4—figure supplement 1a*). Interestingly, these four subtypes in the WD patients exhibited distinct gene expression profiles. Macrophages_3 in CASE displayed elevated expression levels of S100A family genes (*S100A8*, *S100A9*, *S100A12*, *S100A6*, *VCAN*, *FCN1*, and *S100A4*), indicating a potential contribution to MDSC-like macrophages. In contrast, Macrophages_4 in CASE were characterized by upregulation of lipid metabolism genes (*FABP5*, *APOC1*, *CTSD*, and *PLA2G7*) (*Figure 4—figure supplement 1b*; *Zhang et al., 2019a*; *Wang et al., 2022*). By scoring various gene signatures, we compared the M1 polarization, M2 polarization, pro-inflammatory signature, and interferon response signature of four subtypes within two groups (*Figure 4a*). These four subtypes displayed consistently overlapped M1/M2 signatures. Significantly, the Macrophages_3 subtype exhibited a more pronounced pro-inflammatory signal in CASE, while the other subtypes showed a slight decrease. In addition, the interferon response signature showed slight elevation in four subtypes in CASE. Kupffer cells, which are liver-resident macrophages with unique functions in maintaining homeostasis, represent the most abundant population of tissue-resident macrophages in the human body (*Dixon et al., 2013*). KCs were also further clustered and three subtypes were obtained (*Figure 4—figure supplement 1c*). Significant differential expression of genes related to the inflammatory response, including *IL1B*, *CXCL2*, *CCL3C1*, *CXCL8*, and *CCL2*, were identified between the two groups (*Figure 4—figure supplement 1d*). Similarly, KCs subtypes in two groups also showed overlapping M1/M2 signatures, but all subtypes in CASE showed varying degrees of increase in pro-inflammatory and interferon-responsive signatures (*Figure 4b*).

Neutrophils, the most abundant leukocytes in human blood, respond to inflammatory stimuli by migrating to the site of inflammation. Its immunological functions, such as phagocytosis, reactive oxygen species generation, degranulation, and production of cytokines and chemokines, play a crucial role in the resolution of inflammation and tissue repair (*Tang et al., 2021*; *Cho and Szabo, 2021*). In CASE, a lower percentage of neutrophils (3.53% in CASE and 6.45% in CON) was observed, and the functional characteristics were subsequently evaluated. The results showed that the maturation, chemotaxis, phagocytosis, and chemokine activity scores were decreased in CASE, while the activation of the type I interferon signaling pathway was enhanced (*Figure 4c*).

Since the biological process of antigen processing and presentation was upregulated, we calculated the relative expression intensity of genes encoding MHC molecules and complement molecules in cDC1, cDC2, and mature DCs (*Figure 4d*). Notably, genes encoding MHC class I molecules (*HLA-A*, *-B*, *-C*, *-E*, and *-F*) were significantly upregulated in the mature DCs in CASE, while genes encoding MHC class II molecules (*HLA-DQA1*, *-DQA2*, *-DQB1*, *-DQB2*, *-DPA1*, *-DPB1*, *-DRA*, *-DRB1*, and *-DRB5*) were all downregulated in the mature DCs. Besides, genes encoding MHC molecules were not the top significant DEGs in B cells, while the immunoglobulin genes (*IGHD* and *IGHM*) and the naïve B cell marker *TCL1A* were significantly upregulated in CASE (*Figure 4—figure supplement 1e*). Then the immunoglobulin genes (*IGHA1*, *IGHA2*, *IGHG1*, *IGHG2*, *IGHG3*, *IGHG4*, *IGHD*, and *IGHM*) expression was compared in B cells and plasma cells (*Figure 4e*). *IGHA1*, *IGHA2*, *IGHG1*, *IGHG2*, *IGHG3*, and *IGHG4* expression were unchanged in B cells between the two group and all of examined genes were downregulated except *IGHG3* in plasma cells in CASE.

Monocytes accounted for the highest proportion (35.45% in CASE and 41.78% in CON), and most of them were CD14$^+$ classical monocytes (*Figure 4—figure supplement 1f*). Compared to CON,

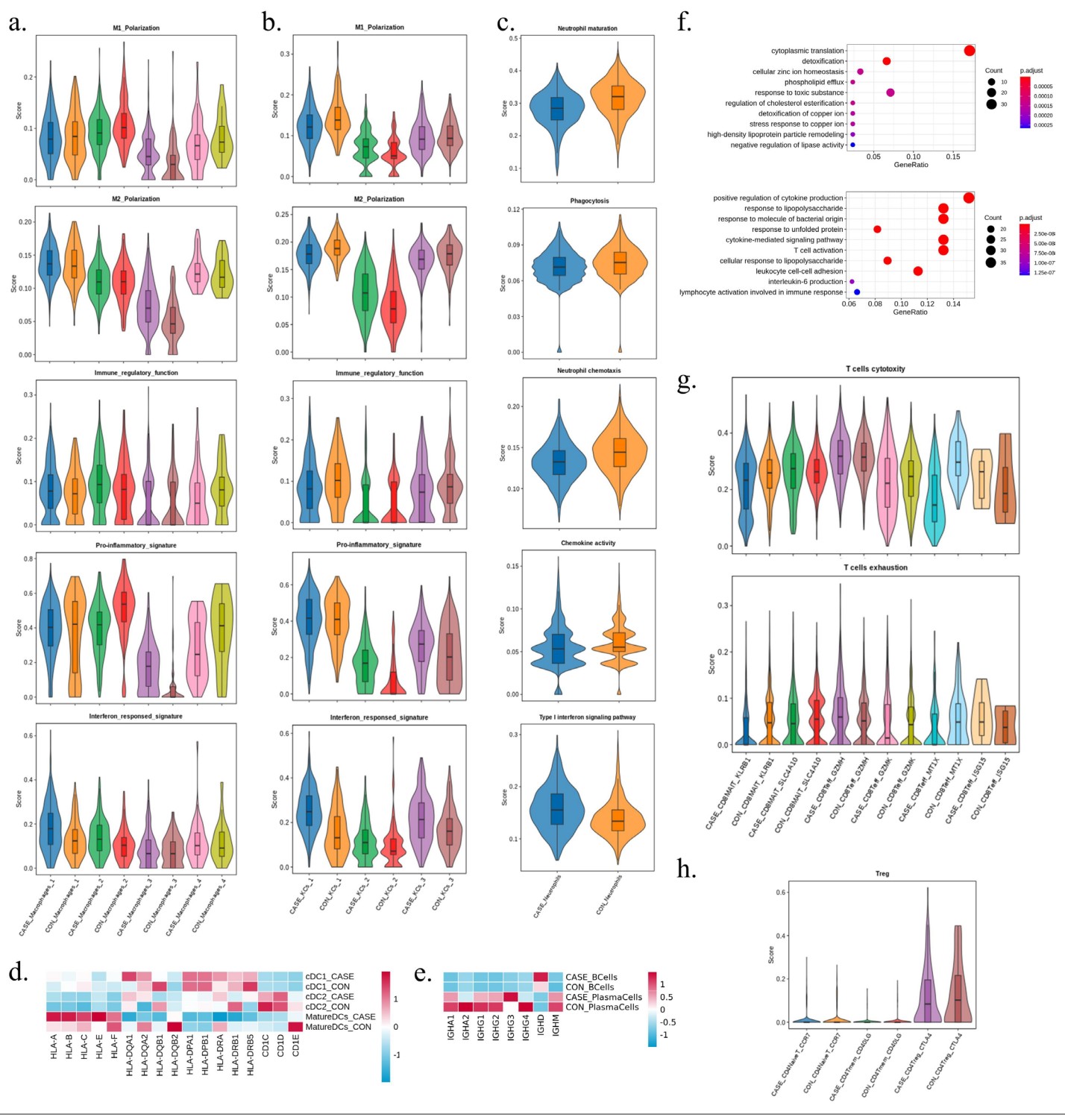

**Figure 4.** The dysfunction of main immune cells in Wilson's disease (WD) patients. (**a**) The violin plots displaying M1 polarization, M2 polarization, pro-inflammatory signature, and interferon responded signature of the four subtypes of macrophages between the two groups by gene scoring. (**b**) The violin plots displaying M1 polarization, M2 polarization, pro-inflammatory signature, and interferon responded signature of the three subtypes of Kupffer cells between the two groups by gene scoring. (**c**) The violin plots displaying maturation, phagocytosis, chemotaxis, chemokine activity, and Type I interferon signaling pathway scores of neutrophils between the two groups by gene scoring. (**d**) The heatmap showing the relative expression level of genes of the three subtypes of DC cells between the two groups. The color represents the relative expression level. (**e**) The heatmap showing the relative expression level of genes of B cells and plasma cells between the two group. The color represents the relative expression level. (**f**) The dot plot showing biological process of Gene Ontology (GO) enrichment on differentially expressed genes (DEGs) in monocytes between the two group. Upper,

*Figure 4 continued on next page*

*Figure 4 continued*

upregulation; down, downregulation. (**g**) The violin plots displaying cytotoxicity and exhaustion scores of the six cell subtypes of CD8 T cells between the two groups by gene scoring. (**h**) The violin plots displaying Treg scores of the three cell subtypes of CD4 T cells between the two groups by gene scoring.

The online version of this article includes the following figure supplement(s) for figure 4:

**Figure supplement 1.** The difference in percentage and gene expression of each cell subtype between CASE and CON.

---

*S100A8*, *S100A9*, and *S100A12* were the top 3 upregulated genes in classical monocytes of CASE (*Figure 4—figure supplement 1g*). *S100A8* and *S100A9* are confirmed to play vital roles in modulating the inflammatory responses by recruiting leukocytes and inducing cytokine secretion during inflammation, and as an alarmin, their elevation indicates inflammation in the local environment (*Wang et al., 2018*; *Chen et al., 2023*). GO analysis of DEGs in monocytes showed a decrease in biological processes involving cytokine production, response to lipopolysaccharide and bacteria, as well as lymphocyte activation (*Figure 4f*).

For T cells, we performed GO analysis on DEGs in CD8 T cells between two groups. Notably, the biological processes linked to energy and lipid metabolism were upregulated in several subtypes, including CD4Tmem_CD40LG, CD4Treg_CTLA4, CD8MAIT_KLRB1, CD8MAIT_SLC4A10, CD8Teff_GZMH, and CD8Teff_GZMK. In addition, biological processes associated with response to copper ion were upregulated in the CD8Teff_MT1X subtype (*Figure 4—figure supplement 1h*). Meanwhile, T cell subtypes showed comparable cytotoxicity scores, and there were slight changes between two groups (*Figure 4g*). However, CD8 T cells in CASE exhibited lower levels of exhaustion, except for CD8Teff_GZMH and CD8Teff_ISG15 (*Figure 4g*). It was also discovered that while the proportion of CD4Treg_CTAL4 was increased in CASE, the Treg signature was slightly lower (*Figure 4h*). Therefore, these results suggest that WD patients exhibit variations in immune cell function and certain biological processes.

## NK cell exhaustion in WD patients

The Kyoto Encyclopedia of Genes and Genomes (KEGG) analysis of DEGs in NK cells between CASE and CON revealed downregulation of the NK cell-mediated cytotoxicity, along with the NF-kappa B signaling pathway and chemokine signaling pathway, in CASE (*Figure 5—figure supplement 1a*). Interestingly, GO analysis showed upregulated biological processes of oxidative phosphorylation and ATP metabolic process, implying the altered metabolic program in NK cells and immune microenvironment (*Figure 5—figure supplement 1b*). Since the primary effect of NK cells is cytotoxicity on target cells, we compared the gene expression levels of activating receptors, inhibitory receptors, and effector molecules between two groups. The results showed significantly upregulated expression in *KLRC1* and *TIGIT*, and significantly downregulated expression in *FCGR3A*, *TNF*, *IFNG*, *GZMB*, *GNLY*, and *CCL4* (*Figure 5a*). NK cells were clustered into three NK subtypes, NK_FCER1G, NK_KLRC1, and NK_FCGR3A, by gene expression profile. Analysis of the expression of these genes in three NK subtypes yielded similar results, except for increased expression of *GZMB* and *GNLY* (*Figure 5—figure supplement 1c*). The results were consistent across the three subtypes, suggesting that the results in total NK population were contributed by all three subtypes and not affected by a single composition. Additionally, the proportion of NK cell was higher in CASE, while the proportion of NK_FCER1G and NK_KLRC1 was higher and the proportion of NK_FCGR3A was lower in CASE (*Figure 3b*; *Figure 5—figure supplement 1d*). The respective transcriptome profile was compared in CASE and CON. Consistently, *KLRC1* was top-ranking in all three subtypes among the significantly upregulated DEGs, while *CCL4* and genes encoding heat shock protein were top-ranking in all three subtypes among significantly downregulated DEGs (*Figure 5—figure supplement 1e*). KLRC1 and TIGIT are inhibitory receptors expressed on NK cell surface, and their activation inhibits NK cell functions (*Zhang et al., 2018*; *André et al., 2018*). It was collectively indicated that WD patients have NK cells with exhausted status, which results in poor effector function.

In addition, trajectory analysis showed that NK cells in two groups exhibited a distinct differentiation trajectory. Most of the NK cells in CON differentiated toward cell fate 1, while those in CASE displayed a higher tendency for differentiation toward cell fate 2, with cell grouping occurring at the final stage (*Figure 5—figure supplement 1f*). To ascertain the gene expression profile and the respective cell statuses of the two outcomes, we conducted pseudotime analysis of gene expression in two

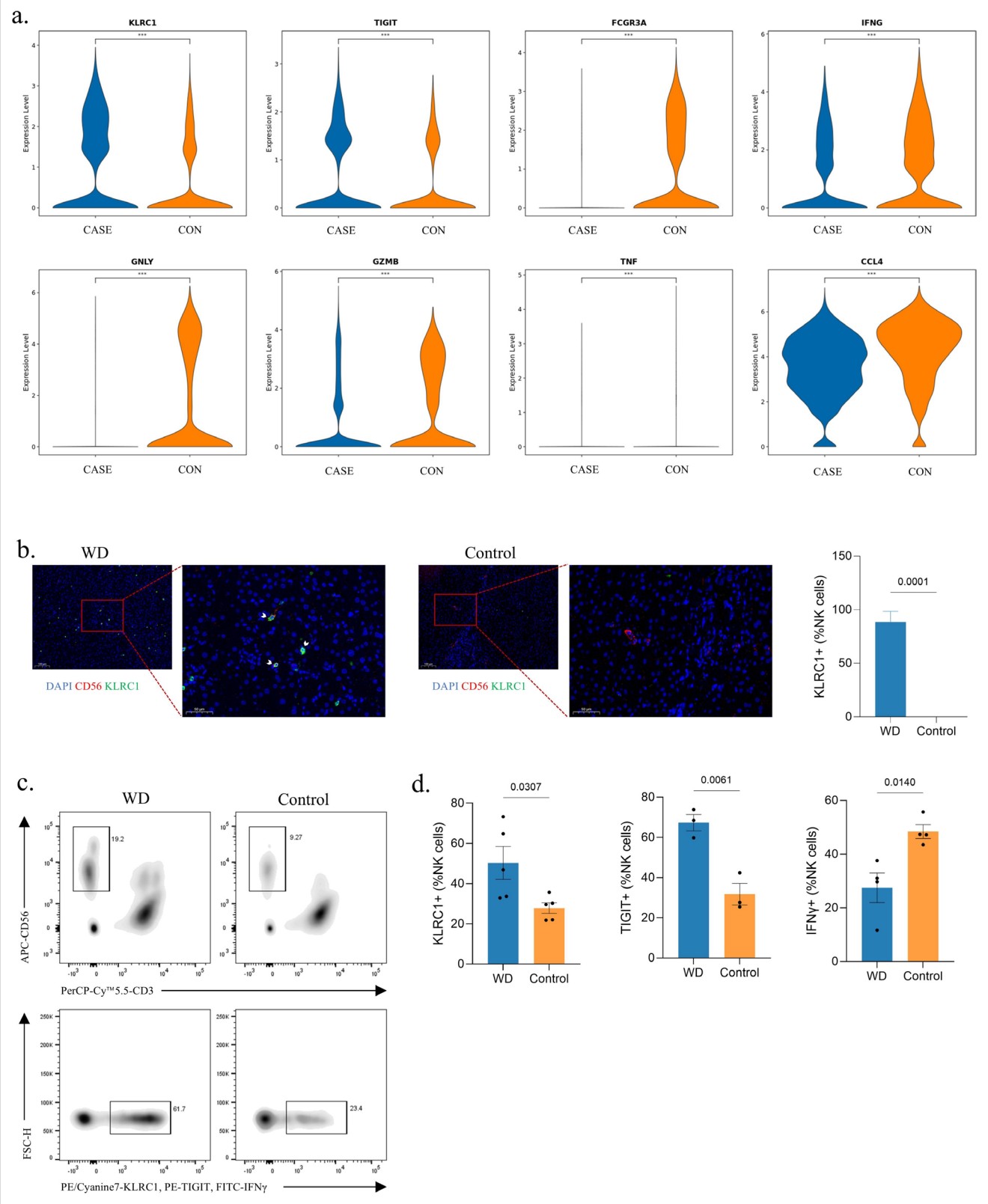

**Figure 5.** Identification of natural killer (NK) cell exhaustion in Wilson's disease (WD) patients. (**a**) The violin plots showing the relative expression level of genes in NK cells between the two group. ***, p<0.001; Wilcox test. (**b**) Representative immunofluorescence staining for NK cells marked by CD56 and expression of KLRC1 in liver tissues of WD and control. The nucleus is stained by DAPI. Scale bars, 100 μm for 40× and 50 μm for 100×. KLRC1⁺ NK cells are highlighted by arrows. The bar chart showing the proportion of KLRC1⁺ NK cells. Quantifications were performed by assessing three random

*Figure 5 continued on next page*

*Figure 5 continued*
fields per slide. Data are mean ± SD. Unpaired two-tailed t-test. (**c**) Gating strategy applied in flow cytometry. Upper: NK cells were determined by CD3⁻ CD56⁺. Down: NK cells with positive indicators (KLRC1, TIGIT, and IFNγ) were recorded. (**d**) The bar charts showing the percentage of KLRC1⁺ (WD, n=5; Control, n=5), TIGIT⁺ (WD, n=3; Control, n=3), and IFNγ⁺ (WD, n=4; Control, n=4) NK cells. Data are mean ± SD. Unpaired two-tailed t-test.

The online version of this article includes the following figure supplement(s) for figure 5:

**Figure supplement 1.** Identification of natural killer (NK) cell exhaustion in Wilson's disease (WD) patients.

differentiating directions. The results showed that the gene expression of *KLRC1* (included in gene cluster 1) was gradually increased along the way to cell fate 2, while the gene expression of *FCGR3A*, *TNF*, and *GZMB* (included in gene cluster 3) was weakened in the same direction, suggesting that cell fate 2 is a more exhausted state of NK cells (*Figure 5—figure supplement 1g*, see full gene list in *Supplementary file 1b*). Consequently, these results demonstrate that NK cells differentiate toward an exhausted status in WD patients. The development, maturation, and function of NK cells are strictly regulated by transcription factors (TFs) (*Brillantes and Beaulieu, 2019*). Next, to assess the TFs underlying differences in NK cells between CASE and CON, we applied PYSCENIC analysis in NK cells. We found that the gene expression of T-bet (encoded by *TBX21*) was downregulated and Eomes (encoded by *EOMES*) was upregulated in CASE (*Figure 5—figure supplement 1h*). These two constitute a pair of TFs that play significantly complementary roles in directing the transcriptional program of NK cells. Consistent with previous studies, T-bet expression was downregulated in exhausted NK cells, whereas Eomes expression was increased (*Sun et al., 2019*; *Zhang et al., 2022*; *da Silva et al., 2014*).

To validate NK cell exhaustion in WD patients, we performed mIHC staining on liver tissue sections and flow cytometry on peripheral blood samples. The mIHC results indicated that KLRC1⁺ cells were more frequently observed in NK cells marked by CD56 in the liver tissue sections of WD patients (*Figure 5b*). Moreover, flow cytometry was also performed to test the number and indicators expression of NK cell from the peripheral blood. After isolation from peripheral blood, NK cells were determined by CD3⁻CD56⁺ and NK cells with positive indicators (KLRC1, TIGIT, and IFNγ) were recorded (*Figure 5c*). The results showed a significant increase in the percentage of KLRC1⁺ and TIGIT⁺ NK cells and a significant decrease in the percentage of IFNγ⁺ NK cells in peripheral blood lymphocytes from WD patients (*Figure 5d*). Meanwhile, the proportion of NK cells in lymphocytes from peripheral blood and the proportion of CD56^Bright NK cells in total NK cells were also calculated. The significantly higher proportion of NK cells in WD patients was consistent with the scRNA-seq data, and more CD56^Bright NK cells suggested a relatively weaker effector function in WD patients (*Figure 5—figure supplement 1i*). Therefore, these results indicated that, despite the increased proportion and the upregulated oxidative phosphorylation, NK cell exhibited exhausted status in WD patients.

## NK cell exhaustion predicts poor prognosis of cholecystitis

Cholecystitis patients with WD exhibit poorer clinical outcomes. There is evidence demonstrating that NK cell exhaustion is correlated with poor liver cancer prognosis (*Sun et al., 2019*; *Sun et al., 2017*; *Sun et al., 2015*). Thus, we hypothesize that NK cell exhaustion also contributes to the worse prognosis of cholecystitis patients with WD.

Prognostic models for inflammatory-related diseases in GEO were developed using machine learning methods, including random survival forest, receiver operating characteristic (ROC) curve, and Kaplan-Meier survival curves. A logistic regression algorithm was used to construct a multigene prediction model based on GSE166915 (cholecystitis) and other inflammatory diseases, namely GSE91035 (pancreatitis), GSE75037 (lung adenocarcinoma), and GSE41804 (hepatocellular carcinoma). Using stepwise regression analysis, 12 genes, *KLRC1*, *TIGIT*, *HAVCR2*, *PDCD1*, *CD96*, *LAG3*, *FCGR3A*, *NCR3*, *NCR2*, *NCR1*, *KLRK1*, and *PRF1*, were selected to obtain the best model. The results showed that the predictive model constructed from these genes had good prognostic performance, with AUC of 0.77, 0.80, 0.97, 0.90 (*Figure 6—figure supplement 1a–d*). The AUCs of the models in GSE75037 and GSE41084 were relatively high, 0.97 and 0.90, respectively.

Univariate Cox regression analysis revealed that the *TIGIT* (in GSE166915) and *PDCD1* (in GSE75037) were risk genes with hazard ratio (HR)>1, while *NCR2* (in GSE91035), *HAVCR2*, *KLRC1*, *NCR3* (in GSE75037), *NCR3*, *LAG3* (in GSE41804) were protective gene with HR<1 among 12 differently expressed markers of NK cell exhaustion (*Figure 6—figure supplement 1a–d*). Consistently, the

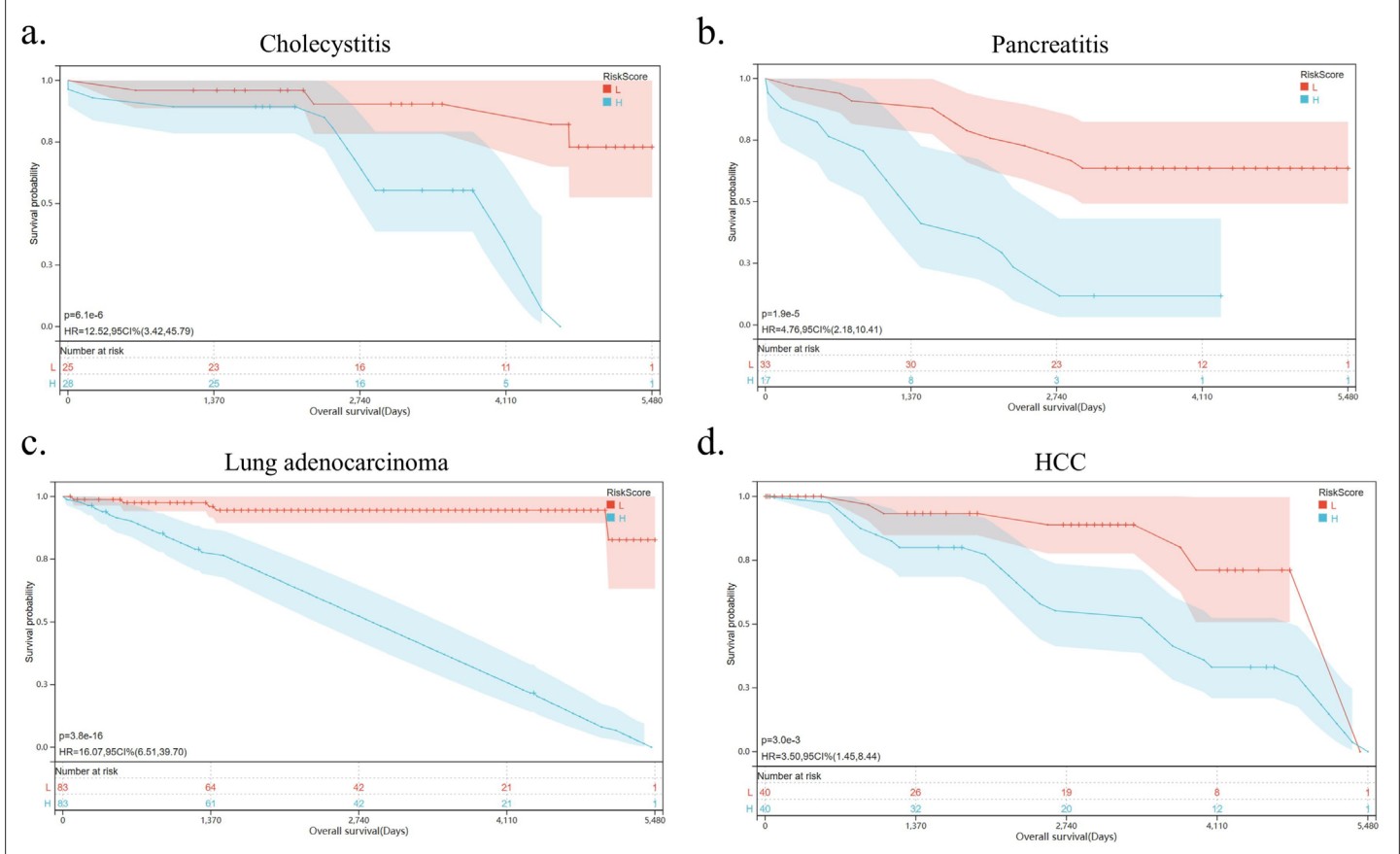

**Figure 6.** Natural killer (NK) cell exhaustion predicts poor prognosis of inflammatory diseases. (**a–d**) Kaplan-Meier survival curves for cholecystitis, pancreatitis, lung adenocarcinoma, and hepatocellular carcinoma according to the markers of NK cell exhaustion in the GSE166915, GSE91035, GSE75037, and GSE41804 datasets.

The online version of this article includes the following figure supplement(s) for figure 6:

**Figure supplement 1.** Predictive model construction for natural killer (NK) cell exhaustion.

Kaplan-Meier survival curves suggested a shorter survival time in the group with high markers of NK cell exhaustion than in the group with low markers in the GSE166915 (p=6.1e-6), GSE91035 (p=1.9e-5), GSE75037 (p=3.8e-16), and GSE41804 (p=3.0e-3) datasets (*Figure 6a–d*). Patients with cholecystitis, pancreatitis, lung adenocarcinoma, and hepatocellular carcinoma who show high markers of NK cell exhaustion are associated with poor survival compared to those with low markers of NK cell exhaustion.

## Discussion

Immune cell dysfunction impacts the development, progression, and prognosis of diseases and serves as the basis for immunotherapy. The current understanding of immune cell dysfunction in inflammatory diseases is limited, which hinders the development of effective therapeutic strategies. Our retrospective clinical analysis indicates that WD contributes to the unfavorable prognosis of cholecystitis. We developed an atlas of immune cell dysfunction linked to specific metabolic disruption in WD patients, including 19 major cell types. The immune microenvironment change primarily encompasses increased proportion and strengthened function in innate immunocytes, including upregulated antigen processing and presentation and activation of immune responses. Furthermore, we conducted a comprehensive assessment and comparison of immune cell function in WD patients and control group. This includes an evaluation of pro-inflammatory and immune regulatory function in macrophages and KCs, phagocytosis and chemokine activity in neutrophils, and cytotoxicity in T cells, etc. We identified NK cell exhaustion in the WD patients, with upregulation of *KLRC1* and *TIGIT*

and downregulation of *CD16*, *TNF*, *IFNG*, *GZMB*, *GNLY*, and *CCL4*. The validation of our findings in patient liver tissues and blood samples was conducted using immunohistochemistry assays and flow cytometry. Our findings indicated that the NK cell exhaustion signature has the ability to predict and prognosticate inflammatory diseases such as cholecystitis, pancreatitis, lung adenocarcinoma, and hepatocellular carcinoma. These results unveil novel perspectives on the immune cell dysfunction role in inflammatory diseases.

WD represents a model of metabolic abnormality in hepatocytes caused by *ATP7B* mutation. Therefore, immune cell dysfunction was induced by the extracellular signals within the reshaped tissue microenvironment. Emerging research indicates that metabolic crosstalk in the tumor microenvironment disrupts metabolic and signaling pathways in immune cells, leading to dysfunction (*Hosseini et al., 2022*; *Park et al., 2023*; *Yan et al., 2023*; *Bader et al., 2020*; *Riera-Domingo et al., 2020*). Targeting metabolic pathways to enhance the efficacy of immunotherapy is a promising approach. Our study found multiple metabolic abnormalities in immune cells, including the widely upregulated ATP metabolic process in T and NK cells. Interestingly, glycolysis and oxidative phosphorylation contribute to maintain the cytotoxic ability and promote expansion of NK cells (*Wu et al., 2020*; *Poznanski and Ashkar, 2019*; *O'Brien and Finlay, 2019*). However, high rate of oxidative phosphorylation, increased proportion, and exhausted status were simultaneously observed in NK cells of WD patients. Further studies are needed to elucidate the metabolic landscape, mechanisms underlying metabolic alterations, and their role in immune cell dysfunction.

## Methods
### Patients and clinical samples
All the clinical data were obtained from The First Affiliated Hospital of Anhui University of Chinese Medicine. The whole blood samples and liver tissue samples were collected from patients at The First Affiliated Hospital of Anhui University of Chinese Medicine. The study was approved by the Ethics Committee of the First Affiliated Hospital of Anhui University of Traditional Chinese Medicine (approval number 2019AH-32), and the clinical study design, informed consent, and case report were also approved in the document.

### Sanger sequencing
The Sanger sequencing on 20 WD patients were conducted by Sangon Biotech (Shanghai) Co., Ltd.
The primers to detect mutations are:

R778L-F, 5'-AGCCTTCACTGTCCTTGTCTTTC-3';
R778L-R, 5'-ATTAGCTGGGATTTCAGAAGTAGTG-3';
T935M-F, 5'-ATGCTTGTGGTGTTTTATTTCTTC-3';
T935M-R, 5'-AGAAGCAAGCAAATAAAATGTAATG-3';
P992L-F, 5'-TCTCAACCTGCCTCTGACTCTG-3';
P992L-R, 5'-TGGTGGCTACTCTGTTGCTACTG-3'.

### Cell culture, gene knockout, and metabolic assays
WRL 68 cells (purchased from Guangzhou Cellcook Biotech Co., Ltd.) were cultured using DMEM high-glucose media supplemented with 10% FBS at 37°C with 5% $CO_2$. The lentivirus-mediated CRISPR-Cas9 KO vector targeting human *ATP7B* was obtained from OBiO Technology (Shanghai) Co., Ltd. Cells at 30–40% confluency were incubated in a medium containing optimal dilutions of lentivirus, then subjected to puromycin selection to obtain the stable transfected cells. OCRs and ECARs were measured using Seahorse XF Cell Mito Stress Test Kit (103015-100, Agilent) and Seahorse XF Glycolysis Stress Test Kit (103020-100, Agilent) according to manuals in Agilent XFe96 Analyzer, respectively.

### Single-cell RNA-sequencing
In the cell preparation section: For the quality check and counting of single-cell suspension, the cell survival rate is generally above 80%. The cells that have passed the test are washed and resuspended to prepare a suitable cell concentration of 700–1200 cells/μL for 10x Genomics Chromium. The system is operated on the machine.

Steps in GEM Creation and Thermal Cycling: GEMs (Gel Bead in Emulsion) were constructed for single-cell separation according to the number of cells to be harvested. After GEMs were normally formed, GEMs were collected for reverse transcription in a PCR machine for labeling.

Post Cycling Cleanup and cDNA Amplication: The GEMs were oil-treated, and the amplified cDNA was purified by magnetic beads, and then subjected to cDNA amplification and quality inspection.

Library Preparation and Quantification: The 3'Gene Expression Library was constructed with the quality-qualified cDNA. After fragmentation, adaptor ligation, sample index PCR, etc., the library is finally quantitatively examined.

Sequencing: The final library pool was sequenced on the Illumina HiSeq instrument using 150-base-pair paired-end reads.

## Pathway enrichment analysis

To investigate the potential functions of DEGs, the GO and KEGG analysis were used with the 'cluster-Profiler' R package 3.16.1 (*Yu et al., 2012*). Pathways with p_adj value less than 0.05 were considered as significantly enriched. GO gene sets including molecular function, biological process, and cellular component categories were used as reference.

## Cell-cell interaction analysis

CellChat (version 0.0.2) was used to analyze the intercellular communication networks from scRNA-seq data (*Jin et al., 2021*). A CellChat object was created using the R package process. Cell information was added into the meta slot of the object. The ligand-receptor interaction database was set, and the matching receptor inference calculation was performed.

## Trajectory analysis

Cell differentiation trajectory was reconstructed with Monocle2 (*Qiu et al., 2017*). Highly variable genes were used to sort cells in order of spatial-temporal differentiation. We used DDRTree to perform FindVairableFeatures and dimension reduction. Finally, the trajectory was visualized by plot_cell_trajectory function. Next, CytoTRACE (a computational method that predicts the differentiation state of cells from scRNA-seq data using gene Counts and Expression) was used to predict the differentiation potential (*Gulati et al., 2020*).

## TF regulatory network analysis

TF network was constructed by pyscenic v0.11.0 using scRNA expression matrix and TFs in AnimalTFDB (*Van de Sande et al., 2020*). First, GRNBoost2 predicted a regulatory network based on the co-expression of regulators and targets. CisTarget was then applied to exclude indirect targets and to search TF binding motifs. After that, AUCell was used for regulon activity quantification for every cell.

## UCell gene set scoring

Gene set scoring was performed using the R package UCell v 1.1.0 (*Andreatta and Carmona, 2021*). UCell scores are based on the Mann-Whitney U statistic by ranking query genes in order of their expression levels in individual cells. Because UCell is a rank-based scoring method, it is suitable to be used in large datasets containing multiple samples and batches. The gene sets for signature scoring are listed in *Supplementary file 1c*.

## Cell type annotation and DEGs analysis

The cell type identity of each cluster was determined with the expression of canonical markers found in the DEGs using SynEcoSys database. The markers and references are shown in *Supplementary file 1d*. To identify DEGs, we used the Seurat FindMarkers function based on Wilcox likelihood-ratio test with default parameters, and selected the genes expressed in more than 10% of the cells in a cluster and with an average log (Fold Change) value greater than 0.25 as DEGs. For the cell type annotation of each cluster, we combined the expression of canonical markers found in the DEGs with knowledge from literature, and displayed the expression of markers of each cell type with heatmaps that were generated with Seurat Heatmap function. Doublet cells were identified as expressing markers for different cell types, and removed manually.

## Multiplex immunohistochemistry

The procedure of deparaffinization, dehydration, antigen retrieval, endogenous peroxidases quench, block, antibody incubation was conducted as standard immunohistochemistry. Solution of tyramide signal amplification (TSA) (WAS1101010, BioMed World) was applied on the slides and antibodies were removed by microwave heating. Repeat block, antibody incubation, TSA, and microwave heating to perform next cycle of staining. DAPI (WAS1301050, BioMed World) was used to stain nuclei and anti-fade mountant (WAS1302050, BioMed World) was applied. The primary antibody to CD56 (ab75813, Abcam) and KLRC1 (ab260035, Abcam) and secondary antibody (WAS1201100, BioMed World) were used. Image of immunofluorescence-stained slides were acquired with Nikon Eclipse Ci-L and 3DHIS-TECH Pannoramic MIDI II.

## Flow cytometry

Lymphocytes were isolated from the whole blood using density gradient centrifugal method (P8610, Solarbio). Isolated cells were blocked (564219, BD Pharmingen) and stained with antibodies against CD3 (560835, BD Pharmingen), CD56 (555518, BD Pharmingen), KLRC1 (375114, BioLegend), and TIGIT (568672, BD Pharmingen). For intracellular cytokine staining, cells were stimulated for 4 hr with Leukocyte Activation Cocktail and BD GolgiPlug (550583, BD Pharmingen) according to manuals. After stimulation, cells were blocked, stained for surface markers, fixed, and permeabilized with Fixation/Permeablization Kit (554714, BD Pharmingen). Fixed cells were stained with antibodies against IFNγ (552887, BD Pharmingen).

## Construction and verification of prognostic model

On the basis of the markers of NK cell exhaustion, univariate Cox analysis was conducted in chole-cystitis, pancreatitis, lung adenocarcinoma, and hepatocellular carcinoma patients to screen survival-related markers of NK cell exhaustion, and markers with p-values<0.05 were retained. We conducted multivariate analysis to identify the optimal prognostic markers for the prognostic model. The risk scores of the patients were calculated based on the normalized gene expression levels and the Cox regression coefficients of selected markers. Time-dependent ROC curves were utilized to verify the prognostic performance of the model for overall survival. For Kaplan-Meier survival, we used the R software package maxstat (maximally selected rank statistics with multiple p-value approximations version: 0.7–25) to calculate the optimal cutoff value for RiskScore. We set the minimum group sample size to be greater than 25% and the maximum group sample size to be less than 75%, ultimately obtaining the optimal cutoff value. Based on this, patients were divided into high and low groups, and further analyzed the prognostic differences between the two groups using the R software package's survival function. The logrank test method was used to evaluate the significance of prognostic differences between different groups of samples, and significant prognostic differences were ultimately observed.

## Statistical analysis

Statistical analyses were performed using appropriate tests as indicated in legends (unpaired two-tailed t-test, chi-square test, and Wilcox test).

## Acknowledgements

This work was supported by grants from National Natural Science foundation of China (Project: 2021KY340; NSFC: 81773112 and 82073186 for QF). This work was supported by Construction project of the State Administration of Traditional Chinese Medicine in 2021 (Teaching Letter [2021] No. 270 for QY) and Anhui Province Clinical Key Specialty Project in 2022 (Anhui Weichuan [2022] No. 297 for QY).

## Additional information

### Funding

| Funder | Grant reference number | Author |
|---|---|---|
| National Natural Science Foundation of China | 2021KY340 | Qiyu Feng |
| National Natural Science Foundation of China | 81773112 | Qiyu Feng |
| National Natural Science Foundation of China | 82073186 | Qiyu Feng |
| State Administration of Traditional Chinese Medicine of the People's Republic of China | Teaching Letter [2021] No. 270 | Qingsheng Yu |

The funders had no role in study design, data collection and interpretation, or the decision to submit the work for publication.

### Author contributions

Yong Jin, Data curation, Validation, Visualization, Methodology, Writing - original draft, Writing – review and editing; Jiayu Xing, Data curation, Validation, Visualization, Methodology, Writing – review and editing; Chenyu Dai, Data curation, Software, Visualization, Methodology, Writing - original draft; Lei Jin, Resources, Data curation, Validation; Wanying Zhang, Methodology, Writing – review and editing; Qianqian Tao, Ziyi Li, Resources, Methodology; Mei Hou, Validation, Methodology; Wen Yang, Conceptualization, Supervision, Writing – review and editing; Qiyu Feng, Conceptualization, Supervision, Funding acquisition, Writing – review and editing; Hongyang Wang, Conceptualization, Supervision; Qingsheng Yu, Conceptualization, Supervision, Funding acquisition

### Author ORCIDs

Qiyu Feng http://orcid.org/0009-0004-5032-7500
Qingsheng Yu https://orcid.org/0000-0002-8074-0329

### Ethics

The study was approved by the Ethics Committee of the First Affiliated Hospital of Anhui University of Traditional Chinese Medicine (2019AH-32.32), and the clinical study design, informed consent, and case report were also approved in the document.

Reviewer #2 (Public review): https://doi.org/10.7554/eLife.98867.3.sa1
Author response https://doi.org/10.7554/eLife.98867.3.sa2

## Additional files

### Supplementary files

Supplementary file 1. Supplementary informations for data analysis. (a) The number and proportion of each cell type. This table shows the number and proportion of each cell type and each subtype in all samples. (b) The list of genes in each cluster. This table shows the gene ID included in three clusters, and the genes mentioned in the article are marked in yellow. (c) The list of genes used for signature scoring. This table shows the gene ID used for signature scoring. (d) The list of marker genes used for cell type annotation. This table shows the marker genes used for cell type and subtype annotation and the respective references.

MDAR checklist

### Data availability

sc-RNA sequencing data that support the findings of this study have been deposited in Gene Expression Omnibus (GEO) with the accession number GSE254082.

The following dataset was generated:

| Author(s) | Year | Dataset title | Dataset URL | Database and Identifier |
|---|---|---|---|---|
| Jin J, Xing J, Dai C, Jin L, Zhang W, Tao Q, Hou M, Li Z, Yang W, Feng Q, Wang H, Yu Q | 2024 | NK Cell Exhaustion in Wilson's Disease Revealed by Single-cell RNA Sequencing Predicts the Prognosis of Cholecystitis | https://www.ncbi.nlm.nih.gov/bioproject/?term=GSE254082 | NCBI BioProject, GSE254082 |

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
