## [Editor Report · eLife Assessment]

This study presents **valuable** findings, based on **solid** methods, to link metabolic dysfunction in Wilson's disease to immune cell dysregulation and poor cholecystitis outcomes. The integration of clinical data and single-cell analyses highlights NK cell exhaustion as a key factor, offering insights with potential therapeutic implications. The work will be of interest to colleagues in inflammatory and metabolic diseases.

---

## [Referee Report · Reviewer #2 (Public review)]

Summary:

Wilson's disease is a rare genetic disorder caused by mutations in the ATP7B gene. Previous studies have documented that ATP7B mutations can disrupt copper metabolism, affecting brain and liver function. In this paper, the authors performed a retrospective clinical study and found that Wilson's disease has a high incidence of cholecystitis. Single-cell RNA-seq analysis revealed changes in the immune microenvironment, including the activation of immune responses and the exhaustion of natural killer cells.

Strengths:

A key finding of this study is that the predominant ATP7B gene mutation in the Chinese population is the 2333G>T (p. R778L) mutation. The authors reported associations between Wilson's disease and cholecystitis, as well as the exhaustion of natural killer cells.

Weaknesses:

The underlying mechanisms linking ATP7B mutations to cholecystitis and natural killer cell exhaustion remain unclear. Specifically, it is not yet determined whether copper metabolism alterations directly cause cholecystitis and natural killer cell exhaustion, or if these effects are secondary to liver dysfunction.

Comments on revisions:

The authors fully addressed my questions and I don't have further comments.

---

## [Author Response]

The following is the authors’ response to the original reviews.

**Reviewer #1 (Public Review):**
Summary:Wilson's Disease (WD) is an inherited rare pathological condition due to a mutation in ATP7B that alters mitochondrial structure and dysfunction. Additionally, WD results in dysregulated copper metabolism in patients. These metabolic abnormalities affect the functions of the liver and can result in cholecystitis. Understanding the immune component and its contribution to WD and cholecystitis has been challenging. In this work, the authors have performed single-cell RNA sequencing of mesenchymal tissue from three WD patients and three liver hemangioma patients.Strengths:The authors describe the transcriptomic alterations in myeloid and lymphoid compartments.Weaknesses:In brief, this manuscript lacks a clear focus, and the writing needs vast improvement. Figures lack details (or are misrepresented), the results section only catalogs observations, and the discussion needs to focus on their findings' mechanistic and functional relevance. The major weakness of this manuscript is that the authors do not provide a mechanistic link between the absence of ATP7B and NK cells' impaired/altered functions. While the work is of high clinical relevance, there are various areas that could be improved.

In this study, we reported for the first time that *ATP7B* mutation and the resulting metabolic abnormalities in hepatocytes cause functional alteration of immune cells in WD patients. We dissected the transcriptional profiles of liver mesenchymal cells and delineated the functional differences of main immune cells in WD patients through scRNA-seq. The NK cell exhaustion and its clinical significance were further demonstrated.

The mechanism study is of our concern. Given that the *ATP7B* mutation is hepatocyte-specific, its effect on immune cells is most probably through intercellular communication rather than through the direct action of ATP7B protein. How *ATP7B* mutation disturbs the metabolic homeostasis in hepatocyte, how metabolic pathways regulate the release of signal substances, and how signal substances act on the NK cells need to be explained. These contents, together with this manuscript, are beyond the scope of a single article, so we put the novelty in this manuscript.

We sincerely appreciate the comments. We have improved the manuscript based on your valuable suggestions. The mechanism study is our subsequent research topic. We are actively promoting it and have found that *ATP7B* mutation rewires a certain metabolism pathway in hepatocyte, and that a critical metabolite functions as the mediator causing NK cell exhaustion.

**Reviewer #2 (Public Review):**
Summary:Wilson's disease is a rare genetic disorder caused by mutations in the ATP7B gene. Previous studies have documented that ATP7B mutations can disrupt copper metabolism, affecting brain and liver function. In this paper, the authors performed a retrospective clinical study and found that Wilson's disease has a high incidence of cholecystitis. Single-cell RNA-seq analysis revealed changes in the immune microenvironment, including the activation of immune responses and the exhaustion of natural killer cells.Strengths:A key finding of this study is that the predominant ATP7B gene mutation in the Chinese population is the 2333G>T (p. R778L) mutation. The authors reported associations between Wilson's disease and cholecystitis, as well as the exhaustion of natural killer cells.Weaknesses:

The underlying mechanisms linking ATP7B mutations to cholecystitis and natural killer cell exhaustion remain unclear. Specifically, it is not yet determined whether copper metabolism alterations directly cause cholecystitis and natural killer cell exhaustion, or if these effects are secondary to liver dysfunction.

In this study, we reported for the first time that *ATP7B* mutation and the resulting metabolic abnormalities in hepatocytes cause functional alteration of immune cells in WD patients. We dissected the transcriptional profiles of liver mesenchymal cells and delineated the functional differences of main immune cells in WD patients through scRNA-seq, focusing on the NK cell exhaustion and its clinical significance.

The mechanism study is of our concern. Given that the *ATP7B* mutation is hepatocyte-specific, its effect on immune cells is most probably through intercellular communication, so we prioritize the studying of this aspect. How *ATP7B* mutation disturbs the metabolic homeostasis in hepatocyte, how metabolic pathways regulate the release of signal substances, and how signal substances act on the NK cells need to be explained. These contents, together with this manuscript, are beyond the scope of a single article, so we put the novelty in this manuscript.

We sincerely appreciate the comments. The mechanism study is the topic of our follow-up study. We are actively promoting the research and we have found that *ATP7B* mutation rewires a certain metabolism pathway in hepatocyte, and that a critical metabolite functions as the mediator causing NK cell exhaustion.

**Reviewer #1 (Recommendations For The Authors):**
Major:(1) Abstract. A major portion of this manuscript focuses on non-NK cells. Data that describes NK cell exhaustion is only minimal. Therefore, the authors should modify the abstract.

Thank you for your valuable suggestion. We have supplemented the description of functional changes in other immune cells, and have modified the abstract (line 31-35).

(2) Introduction. There are three paragraphs. The first paragraph discusses cholecystitis. However, there are too many repetitions, and the information is unclear. In the second part, the authors discuss NK cells and their exhaustion. The authors do not establish a clear rationale or logic linking NK cells to WD or cholecystitis. In the last paragraph, the authors describe their findings. Their correlation between NK cell exhaustion and the poor healing process of cholecystitis has no direct experimental proof.

Thank you for your comments. We have deleted the repetitions and rephrased some sentences (line 72-74). Briefly, in the first paragraph, we proposed the significant prognostic value of immune cell dysfunction for cholecystitis. In the second paragraph, we introduced NK cell exhaustion and its potential to predict prognosis of certain diseases. In the third paragraph, we introduced that the liver is a central organ involved in metabolism and immunity, holding a large number of NK cells. Liver pathologies commonly impact the development and outcome of inflammation-associated diseases such as cholecystitis. WD was selected as a research model. In the last paragraph, we introduced our findings from clinical study, scRNA-seq, clinical samples, and bioinformatics analysis, and concluded at the end.

(3) Results. Overall, the results section lacks clarity and a clear focus. Figure legends need to be significantly detailed. The authors make too many broad statements without any support. The authors also make too many overstatements.

Thank you for your valuable suggestion. We have improved the inaccurate statements and made detailed refinement of figure legends. All the changes are marked in the manuscript, and related responses are described below.

Figure 1: No information is provided about the functional impairment of ATP7B protein due to the mutation found in the cohort of Chinese patients. What does 'immune abnormalities' (line 127) mean? What is the relevance of showing liver fibrosis and copper accumulation in the eye in Figure 1c and d, respectively? Total cholesterol concentrations are still within the range in the plasma of WD patients, but the authors call it higher. ECAR has not changed in WD patients, but the authors claim it has (line 117).

(1) All these gene mutations in WD disable the protein function and cause the same outcome. (2) We have deleted the inappropriate statement. (3) In clinical observation, we found that WD not only causes copper accumulation in hepatocytes, but also leads to a variety of diseases, including liver fibrosis, Kayser-Fleischer Ring, and lower risk of hyperglycemia. We showed these together with the data of cholecystitis incidence. We think these might suggest the significance of intercellular communication between hepatocytes and other cells in microenvironment. (4) We have deleted the inappropriate statement (line 108-110, 112-113).

Figure 2: Did the authors use the liver mesenchymal tissue or mesenchymal cells? Figure 2 states that they used mesenchymal cells, different from liver mesenchymal tissue. Numbers within Figure 2b UMAP are not visible. Were the initial T and NK cells annotated as indicated in Figure S2 (CD3D, CD#E, CD3G)? If so, that does not include NK cells.

(1) The liver mesenchymal cells were used for scRNA-seq. (2) It is possible that the image resolution was reduced due to the compression of files by the submission system during merging process. We confirm that the image resolution of all figures meets publishing requirements, and that all characters on the figures are visible. You can download figure files to view details. (3) It was our negligence that the incomplete cell markers were shown in Figure S2. We have updated the markers (CD3D, CD3E, NKG7), references (Ref #53, #55, and #56), and related figures (Figure 2e, and Figure S2c).

Figure 3: The authors should change 'Case' to 'WD patients' both in the text and figures. DEGs in Figure 3C indicate a transcriptomic alteration in the B cell compartment, which the authors do not delineate. Also, the rationale and explanation for the CellChat analyses are minimal. Concluding that a change occurred within the TME with minimal data and explanations is unfair.

Thank you for your comments. (1) We apologize for the confusion caused by the use of nomenclatures and abbreviations in the text and figures. In all scRNA-seq data analysis, presentation, and description, we used specific terms (CASE and CON) to refer to the group of WD patients and controls, as well as their cell population. We have now unified the use of nomenclature in full text and defined them when first appeared (line 126-127), avoiding using lowercase form to prevent confusion. (2) We have now compared the expression of key genes of B cell between the two group in the next section “The dysfunction of main immune cells in WD patients” (line 230-235, Figure 4e, Figure S4e). (3) We have described the results of cellular communication in more detail (line 188-194). (4) We have modified the conclusion and all the related statement in full text (line 29-31, 82-84, 149, 194-195).

Figure 4: This section deals with multiple cell types with minimal explanations. This section discusses various cell types, but it lacks focus. In particular, the T cell section should be separated and elaborated more in detail.

(1) In this section, we intended to show the comparison in function of main immune cells that account for a considerable proportion, instead of just showing differently expressed genes that provide minimal information. The evaluation of functional signature, based on the integration of multiple gene expression, allows a direct understanding of the final outcome owing to transcriptional changes. (2) Given that the main functions of T cells did not change significantly and there were more significant changes in innate immunity, the T cell section is relatively short and unsuitable as a separated part.

Figure 5: What are the distinct subsets of NK cells authors have found in the WD patients and controls? How do these subsets differ between the two groups in numbers and their transcriptomes? The presentation and labeling of Figure 5 and Supplementary Figure 5 need to be vastly improved. The pseudotime presentation in Figure 5b should be presented separately for the patients and the controls. Are the changes in gene expression presented in Figure 5a due to the change in the subset compositions? Figure 5c immuno-staining is not at all visible. A clear explanation should be given for the differences between Figure 5c and Figure 5e, where NKG2A expressions are shown. A better explanation for Figure 5d is required. Did the authors use all the antibodies with the same fluorochrome? If so, what color is that? Can the authors include the individual samples in the bar diagram in Figure 5e? Again, the data in Figure 5 is insufficient to conclude that NK cells are exhausted in WD patients. While the role of changes in the expression of T-BET and EOMES can be related to dysfunction and cellular exhaustion of NK cells, the statement made by the authors needs to be toned down as they do not test with independent experiments.

(1) The subsets of NK cell were clustered by gene expression profile and labeled by the characteristically expressed gene, using certain algorithm in the routine procedure. They cannot be distinguished in clinical samples by one or several genes or other sorting methods. Thus, we were not able to analyze these subsets in clinical samples. (2) We have supplemented the comparison of numbers and transcriptomes of three NK subtypes between the two groups (line 268-273). (3) We have checked the figures and confirmed that all characters on the figures are visible. (4) We have separately presented the plot in Figure S5d. (5) We compared the expression level of genes presented in Figure 5a between the two groups in three NK subtypes and supplemented this part (line 264-268). The results were very consistent across the three subtypes, suggesting that the results in total NK population were contributed by all three subtypes and not affected by a single composition. (6) KLRC1 is also known as NKG2A. We are sorry for not making a clear explanation, and now we use KLRC1 only in all text to avoid confusion. We have made a more clear and detailed description for Figure 5c, 5d, and 5e (now labeled as Figure 5b, 5c, and 5d), and have included the fluorochrome in Figure 5d (now labeled as Figure 5c) and the individual value in Figure 5e (now labeled as Figure 5d) (line 293-299). (7) In this section, we found the upregulated expression of inhibitory receptors, downregulated expression of effector molecules, and the impaired NK cell-mediated cytotoxicity in NK cell of WD patients from scRNA-seq. Then we validated the findings in clinical liver section samples and clinical blood samples by mIHC and flow cytometry, respectively. According to the recent articles, exhausted NK cells are characterized by decreased production of effector cytokines (e.g., IFNγ), as well as by impaired cytolytic activity, and downregulate expression of certain activating receptors and upregulate expression of inhibitory receptors (e.g., 10.3389/fimmu.2017.00760, 10.1038/s41590-018-0132-0, 10.1038/s41467-019-09212-y, 10.1080/2162402X.2016.1264562). Therefore, we concluded NK cell exhaustion in WD patients. (8) In the part about transcription factors, we kept the description of objective data and deleted the statement of the contribution of transcription factors to NK exhaustion.

Figure 6: Data presented in Figure 6 and the conclusion made in this manuscript are predictive. There is no direct testing of ATP7B in NK cells to show the functions of this gene. Extension of this to patient survival is purely speculative. As long as authors state these facts clearly in their text, it can be acceptable. However, they do not extend their conclusions to similar liver diseases.

*ATP7B* mutation is hepatocyte-specific, and it does not occur in any immune cells. The function of ATP7B in NK cell was not studied. We found the NK exhaustion and poor prognosis of cholecystitis in WD patients. Given that there were researches demonstrating that NK exhaustion is correlated with poor liver cancer prognosis, we hypothesized that NK exhaustion contributes to the poor prognosis of cholecystitis. Bioinformatics studies confirmed our hypothesis and supported the extension of this result to other inflammatory diseases. We had no experimental data, but this result was reliable in bioinformatics method.

(4) Discussion: While the authors analyzed multiple cell types, the discussion is primarily focused on NK cells. There is no clear link between copper utilization, NK cell function, and exhaustion that the authors articulate.

Thank you for your comments. The focus of our study is NK cell exhaustion, which is experimentally proven, so we discussed this aspect. We prioritize the effect of intercellular communication and metabolic alteration on the NK cell exhaustion in our follow-up study. Excess copper is released into the circulation in some circumstances in WD patients, but generally they receive long-term de-coppering therapy to maintain intracellular copper at a non-lethal level. Thus, we do not tend to consider copper as a critical factor in this study. In original manuscript, we mentioned the cuproptosis and its potential as a novel target. It is likely to lead to ambiguity and misunderstanding, so we deleted this part to put our point of view clearly.

(5) Supplementary Figures: The presentation and labeling of these figures need to be changed.

Thank you for your suggestions. We have modified the figures and confirmed that all characters on the figures are visible.

**Reviewer #2 (Recommendations For The Authors):**
It is better to test whether ATP7B mutation can directly affect immune functions.

Thank you for your suggestions. Given that the *ATP7B* mutation is hepatocyte-specific, its effect on immune cells is most probably through intercellular communication. Thus, we prioritize the effect of intercellular communication on the NK cell exhaustion and we are actively promoting the research.